# Antimicrobial Management of Skin and Soft Tissue Infections among Surgical Wards in South Africa: Findings and Implications

**DOI:** 10.3390/antibiotics12020275

**Published:** 2023-01-31

**Authors:** Atlanta B. Makwela, Wandisile M. Grootboom, Veena Abraham, Bwalya Witika, Brian Godman, Phumzile P. Skosana

**Affiliations:** 1Department of Clinical Pharmacy, School of Pharmacy, Sefako Makgatho Health Sciences University, Molotlegi Street, Ga-Rankuwa, Pretoria 0208, South Africa; 2Dr George Mukhari Academic Hospital, Molotlegi Street, Ga-Rankuwa, Pretoria 0208, South Africa; 3Department of Pharmaceutical Sciences, School of Pharmacy, Sefako Makgatho Health Sciences University, Molotlegi Street, Ga-Rankuwa, Pretoria 0208, South Africa; 4Department of Public Health Pharmacy and Management, School of Pharmacy, Sefako Makgatho Health Sciences University, Pretoria 0208, South Africa; 5Strathclyde Institute of Pharmacy and Biomedical Sciences, University of Strathclyde, Glasgow G4 0RE, UK; 6Centre of Medical and Bio-allied Health Sciences Research, Ajman University, Ajman P.O. Box 346, United Arab Emirates

**Keywords:** antimicrobial utilization, antimicrobial resistance, guidelines, hospitals, post-surgical care, skin and soft tissue infections, South Africa

## Abstract

Skin and soft tissue infections (SSTIs) are one of the most common infectious diseases requiring antibiotics. However, complications of SSTIs may lead to the overprescribing of antibiotics and to subsequent antibiotic resistance. Consequently, monitoring the prescribing alignment with the current recommendations from the South African Standard Treatment Guidelines (STG) is necessary in order to improve future care. This study involved reviewing pertinent patients with SSTIs who were prescribed antimicrobials in the surgical ward of a leading South African tertiary public hospital from April to June 2021 using an adapted data collection tool. Sixty-seven patient files were reviewed. Among the patients with SSTIs, hypertension and chronic osteomyelitis were the most frequent co-morbidities at 22.4% and 13.4%, respectively. The most diagnosed SSTIs were surgical site infections (35.1%), wound site infections (23%), and major abscesses (16.2%). Blood cultures were performed on 40.3% of patients, with *Staphylococcus aureus* (32.7%) and *Enterococcus* spp. (21.2%) being the most cultured pathogens. Cefazolin was prescribed empirically for 46.3% of patients for their SSTIs. In addition, SSTIs were treated with gentamycin, ciprofloxacin, and rifampicin at 17.5%, 11.3%, and 8.8%, respectively, with treatment fully complying with STG recommendations in 55.2% of cases. Overall, the most common cause of SSTIs was *Staphylococcus aureus*, and empiric treatment is recommended as the initial management. Subsequently, culture sensitivities should be performed to enhance adherence to STGs and to improve future care.

## 1. Introduction

Skin and soft tissue infections (SSTI) are common infections with a broad spectrum of presentations and etiologies [1,2,3]. These incorporate superficial infections, which include erysipelas, impetigo, folliculitis, furuncles, and carbuncles located at the epidermal and dermal layers, while cellulitis is located in the dermis and subcutaneous tissues [4,5]. Alongside this, there can be deep infections that extend below the dermis and may involve subcutaneous tissue, fascial planes, or muscular compartments, which present as either complex abscesses, fasciitis, or myonecrosis [5]. These differences have led authors to classify SSTIs as either uncomplicated or complicated [1,5,6,7,8]. Uncomplicated SSTIs are seen as being low risk for life- or limb-threatening infections, unless they are improperly treated, which includes inappropriate antibiotic therapy [9]. However, complicated SSTIs include deep tissue infections and must be treated early to avoid life-threatening infections [1,9]. This includes early culture and sensitivity testing in case of resistant organisms, which are increasing, along with surgical interventions including debridement [5,10,11].

The incidence of SSTIs is increasing, with up to 14 million people presenting with SSTIs each year in the USA, with the number growing, and is most common in those ≥50 years [1,3,12,13,14,15]. In South Africa, the incidence of SSTIs is also increasing [16], with SSTIs now the most common indication for antibiotic use among hospitals across South Africa accounting for 12.97% of patients being prescribed antibiotics [17]. This is lower than 16.2% among the African countries taking part in the Global Point Prevalence Survey; however, again, the most prevalent indication for the prescribing of antibiotics among in-patients [18]. Overall, the incidence rates for SSTIs in the USA in 2009, where there are the most incidence data, was 49 episodes/1000 persons, which is higher than the incidence of pneumonia (4.9 episodes/1000 persons) or urinary tract infections (17.3 episodes/1000 persons) [2]. The increasing incidence of SSTIs is principally associated with aging populations, as well as an increase in the prevalence of metabolic diseases, especially cardiovascular diseases and type 2 diabetes [19]. SSTIs also appear more prevalent among men, and also more prevalent among black people in the USA [20]. Other at-risk populations for SSTIs include parenteral drug users, due to poor injection practices, with 64% of bacterial infections being among people who inject drugs (PWID), and those that are HIV positive, which is particularly important in sub-Saharan Africa with high rates of HIV, including among in-patients [16,17,21,22,23]. As a result of increasing incidence rates, SSTIs are now among the most common causes of hospitalization globally for antibiotic treatment [1,14]. The costs associated with SSTIs are also increasing [24], and were estimated at $13.8 billion in 2012 alone in the USA and rising [25,26]. Consequently, there is an increasing need to treat SSTIs appropriately across countries, including sub-Saharan African countries, to reduce morbidity, mortality, and associated costs.

Most SSTIs are caused by Gram-positive cocci, including *Staphylococcus aureus* and *Streptococci* [3,5,8,24,27,28]. Alongside this, *Pseudomonas aeruginosa, Enterococcus*, and *Escherichia coli* are also seen in hospitalized patients with SSTIs including among burn patients [3,5,8,20]. While bacterial organisms are the main cause of SSTIs, many other organisms may also cause SSTIs, including fungi and mycobacterium [25,29].

The effective management of complicated SSTIs is characterized by surgical debridement, drainage, and antimicrobial therapy [5,30]. Antibiotics remain the mainstay of treatment for SSTIs against causative pathogens [3,5,8,15,30]. However, antibiotic prescribing needs to be monitored due to potential overprescribing by healthcare workers (HCWs). Overprescribing can lead to antimicrobial resistance (AMR), which is increasing worldwide and now represents an appreciable global public health problem [31,32,33,34]. In addition, any antibiotic regimen initially prescribed needs to be rapidly changed if subsequent sensitivity findings show resistance to prevent patient deterioration, especially in more complex cases [3,10,13,30].

In order to ensure that antimicrobials for SSTIs are used as indicated, it is critical that any agreed national Standard Treatment Guidelines (STGs) are followed and incorporated into ongoing Antimicrobial Stewardship Programs (ASPs) [33,34,35]. This is important as adherence to national STGs is increasingly seen as a marker of good quality care [18,36,37], and continued empiric prescribing in patients with SSTIs is associated with higher costs and mortality [16,38]. ASP activities have been variable across Africa [39]; however, countries such as South Africa provide guidance to other African countries with their proactive measures and examples to enhance antimicrobial stewardship activities [39,40]. Having said this, adherence to published guidelines has been variable in South Africa across the sectors [41,42,43,44], and there has been a low adherence to SSTI guidelines in other countries [45]. ASPs can improve future antimicrobial prescribing, whether part of the activities of the Drug and Therapeutic Committees (DTCs) or the remit of Infection, Prevention, and Control (IPC) committees, especially if there are concerns about DTC activities among African countries [39,46,47,48,49].

Consequently, we believe there is a need to investigate the antimicrobial management of SSTIs in a leading tertiary hospital in South Africa, alongside adherence to the current South African STG. We are aware that studies have been undertaken to assess the management of SSTIs in the Emergency Departments of hospitals in South Africa, as well as the outcomes thereof; however, adherence to current guidelines was not assessed in these studies [16]. We are also aware that SSTIs are a common reason for the prescribing of antibiotics in hospitals in South Africa [17]. In view of this, we wanted to concentrate on in-patients with SSTIs, as these tend to be more severe and more complicated, thereby increasing costs [16,50,51]. The findings can be used to guide future prescribing practices in South Africa as part of potential ASPs as the authorities in South Africa seek to reduce AMR as part of the agreed national action plan [34,52,53].

## 2. Results

### 2.1. Demographic and Clinical Characteristics of the Patients

A total of 67 patient files from the surgical wards were reviewed over a 3-month period between 1 April and 30 June 2021. The demographics of the patients involved in this study are depicted in Table 1.

The majority of the patients’ files reviewed were males (47; 70.1%), with most patients presenting with co-morbidities. Co-morbidities included hypertension, chronic osteomyelitis, and HIV, with prevalence rates of 22.4%, 13.4%, and 11.9%, respectively. The predisposing factors for SSTIs were age > 61 (Figure 1), alcohol, smoking, diabetes mellitus, and being male, at 25.4%, 68.7%, 64.2%, 9%, and 70.1%, respectively (Table 1).

Of the total number of patients in this study, 83.6% presented with signs and symptoms of infection, including pus oozing from the infection site and a foul smell, followed by inflammation in 76.4% of cases and skin ulceration in 34.3% of cases. The majority of the patients were newly diagnosed with SSTIs as 77.6% of patients did not have SSTIs before hospital admission.

More patients presented with complicated SSTIs (98.5%) than uncomplicated SSTIs (1.5%). The common complicated SSTIs seen were surgical site infections (SSIs), which accounted for 35.1% of SSTIs, wound infections (23.0%), and major abscesses (16.2%) (Figure 2).

Among the surveyed patients, 15.6% presented with more than one type of complicated SSTIs. Only one patient (1.5%) presented with erysipelas, which is an uncomplicated SSTI, and two patients (3%) presented with other types of skin infections that were not part of the investigated list of SSTIs, i.e., dermatomycosis and dermatitis.

### 2.2. Microbiology Results

Bacterial cultures and antimicrobial sensitivities were conducted in 27 (40.3%) patients. Fifty-two bacterial pathogens of various species were identified in the 27 patients, with the bacterial distribution according to the diagnosis presented in Table 2.

Most of the cultured microorganisms were Gram-positive microorganisms (63.5%) with *Staphylococcus aureus* and *Enterococcus* species accounting for the most at 32.7% and 21.2%, respectively. *Pseudomonas aeruginosa* was the most prevalent Gram-negative microorganism accounting for 12.4% of cases, while *Klebsiella pneumonia* was the least contributing microorganism with only 5.8% prevalence (Figure 3). No anaerobic cultures were seen in the bacterial cultures performed.

The majority of the patients, 14 (59.2%), were infected with more than one bacterial pathogen and, in some cases, patients were infected with both Gram-negative and Gram-positive pathogens.

Most of the Gram-positive microorganisms were isolated from SSIs (27.0%) and skin ulcers (24.2%). The majority of the Gram-negative microorganisms were isolated from wound site infections (26.3%) as well as SSIs (26.3%).

### 2.3. Antibiotic Treatment

A total of 16 different antibiotics were prescribed to patients with SSTIs. Of those patients prescribed antibiotics, 25 (31.3%) received cefazolin, while a further 14 (17.5%) and 9 (11.3%) received gentamycin and ciprofloxacin, respectively (Table 3). Furthermore, 19 (28.4%) patients received a combination of two or more antibiotics as their treatment therapy. Overall, out of the total number of antibiotics prescribed, 66.3% (53/80) were “access” antibiotics, 31.2% (25/80) “watch” antibiotics, and 2.5% (2/80) were “reserve” antibiotics.

*Staphylococcus aureus* was predominantly sensitive to cefazolin (29.0 %) and cloxacillin (29.0%), while *Enterococcus* spp. was sensitive to gentamycin (50.0%) and ciprofloxacin (33.0%). Overall, the majority of microorganisms were found to be sensitive to gentamycin and ciprofloxacin, particularly the Gram-negative microorganisms (Table 4).

### 2.4. Empiric Treatment for Skin and Soft Tissue Infections

Empiric treatment was typically prescribed to patients while waiting for the results of any bacterial culture and susceptibility. Overall, 40 (59.7%) patients continued with empiric treatment, with culture and sensitivity testing only performed in 27 (40.3%) patients.

Four antibiotics were used for empiric treatment (Table 5). Cefazolin was prescribed in 46.3% of patients as empiric treatment; however, it was also given as a definitive treatment in 37.3% of patients.

Fourteen (20.9%) patients did not receive empiric treatment. Instead, they were prescribed targeted antibiotic treatment based on the sensitivity findings.

### 2.5. Other Methods of Treatment

Drainage was the treatment option for patients with abscesses, cellulitis, and leg ulcers, with only 6 (9%) of the patients receiving drainage in our study.

Patients who presented with areas of necrosis or hemorrhage require surgical debridement, and this was performed in 31 (46.3%) of patients.

Wound dressings were performed in 44 (65.7%) of patients presenting with surgical site infections, wound infections, leg ulcers, and cellulitis.

### 2.6. Adherence to Guidelines

After taking into consideration the choice of antibiotics, drainage, debridement, and culture results, it was observed that treatment fully complied with the STG/EML hospital level 5th edition in 55.2% (*n* = 37) of cases. However, there was still a gap of non-compliance in 30 (44.7%) situations. The cases that did not comply with treatment were mainly due to inappropriate antimicrobial choice (34.3%), dosing interval (13.4%), and the lack of prescribing of antibiotics found to be sensitive following microbiology test results (9%).

## 3. Discussion

We believe this is the first study undertaken in a public tertiary hospital in South Africa to fully assess the current management of patients with SSTIs, particularly complicated SSTIs. The most common SSTIs in our study were SSIs, wound infections, and major abscesses. This differs from other studies where abscesses, cellulitis, and diabetic foot ulcers were also common SSTIs alongside SSIs [4,16,28]. We are not sure of the reasons for these differences; however, this may be due to local factors. These include potential co-morbidities with, for instance, concerns with HIV, IV drug use, and diabetes in South Africa and how SSTIs develop, including the behavior of men in the country as well as issues of race [16,22,28].

SSTIs occurred more often in men in our study, similar to other studies [4,13,16,50]. This could be attributed to an increasing tendency among men to engage in more risky behavior, including driving at high speeds, fighting, and using guns in some countries, which increases their risk for injuries with subsequent infections [28]. In our study, men in the surgical wards were admitted due to gunshot wounds, motor vehicle accidents, mob justice due to theft or house breakings, or broken bones due to fights caused by alcohol or illicit substances. South Africa remains one of the top countries for alcohol consumption in Sub-Saharan Africa, alongside a history of violence with either a gun or a knife under the influence of alcohol [54,55]. In 2017 and 2018 in South Africa, 167,352 counts of assaults with the intention of inflicting grievous bodily harm were reported to the authorities [54]. Alongside this, an estimated 73.3% of all robberies in South Africa were perpetrated with weapons including guns and knives, all under the influence of alcohol [54].

The majority of patients in our study presented with hypertension, chronic osteomyelitis, and HIV as comorbidities. This is similar to the findings of Black and Schrock (2018) [16]. The reason for the similarities might be because these conditions result in complications with SSTIs. However, this is different from the findings of Talan et al. (2015), where diabetes was the most common co-morbidity [50]. The differences in the findings between the studies may be due to geographic reasons or race, which need further investigation. The comorbidities found in our study do pose risks for SSTIs, e.g., diabetes mellitus can result in diabetic foot ulcers and chronic osteomyelitis can result in leg ulcers [2,3].

The most prevalent bacterial pathogen in our study was *Staphylococcus aureus*, similar to other studies, with *Staphylococcus aureus* colonizing the skin in approximately 80–90% of all SSTIs worldwide [3,5,8,27,28]. *Enterococcus* spp. was the second most prevalent causative microorganism, similar to a study in Kenya [28]. However, in Kenya, *Proteus* spp. was the least causative microorganism, whereas in this study, *Klebsiella* spp. was the least causative microorganisms [28].

In view of the potential microorganisms involved, published studies have documented appreciable empiric prescribing of the penicillins and cephalosporins, although there can be appreciable prescribing of clindamycin, especially for patients with allergies to the penicillins [16,56]. Vancomycin is also used where methicillin-resistant *Staphylococcus aureus* is suspected [8,45,57], although linezolid is also being prescribed in this situation [8,45,58].

Cefazolin and amoxicillin/clavulanic acid were among the most prescribed antibiotics as empiric treatment in our study. This complies with the South African STG that recommends the use of β-lactams as empiric treatment for SSTIs due to their good activity against Gram-positive microorganisms, particularly *Staphylococcus aureus* and *Streptococcus* spp. [59]. South Africa’s STG also suggests that the treatment of choice for an abscess is drainage and empiric treatment, with the need for antibiotic prescription in only a limited number of situations [59].

The South African STG also suggests that cellulitis, abscesses, or impetigo should be treated with cloxacillin/flucloxacillin or cefazolin as empiric treatment; alternatively, clindamycin if there are allergies to penicillin [59]. However, most of the patients with cellulitis in our study received meropenem, which did not comply with the STG. The reason for the prescribing of meropenem instead of the recommended antibiotics are currently unknown; however, there are possibilities of stock outs necessarily leading to agreed changes [60]. Some patients were also prescribed gentamycin and ciprofloxacin following culture sensitivity assessments. For diabetic foot ulcers and other leg ulcers, amoxicillin/clavulanic acid is recommended in the STG, while cefazolin is recommended for lower limb, pelvic, and amputation surgical infections [59].

Overall, the majority of prescribed treatments in our study complied with the South African STG. However, non-compliance was seen in 44.7% of cases, with similar findings in the study of Ibrahim et al. [56]. These compliance rates were appreciably higher than those seen by Sutton et al. (2020), where only 14% of SSTI patients received guideline-concordant antibiotics [61]; Kamath et al. (2018), where the treatment for SSTIs only complied with guidelines in 20.1% of cases [45]; and Wiltrakis et al., where initial compliance to STGs was only 2% to 28% depending on the SSTI [62]. Low compliance rates could be due to poor practices, a lack of awareness of the guidelines and their content, difficulty in finding them, especially if they are not readily available, or difficulty with following them [45,63].

Non-compliance in our study included the wrong antibiotics being prescribed, a longer treatment duration than recommended, and not prescribing the antibiotics that were found to be sensitive following culture results. Overall, culture and sensitivity testing was only undertaken on 40.3% of occasions in our study. This needs the be addressed going forward as part of ongoing de-escalation strategies [8]. Alongside this, a critical review of the prescribing of “watch” and “reserve” antibiotics over “access” antibiotics needs to be undertaken where pertinent as part of future ASP activities [35,39,64,65]. Having said this, the current target of over 60% “access” antibiotics (66.3%) was achieved [39,66]. Other potential areas to review to reduce unnecessary hospital stay and costs include more rapid switching from IV to oral treatment without compromising care [8,67,68].

Potential ways forward to improve antimicrobial prescribing in this and other hospitals in South Africa are to improve compliance to STGs, which could be via ASPs [8,35,39]. We have seen that ASPs among hospitals can increase compliance to recommended antibiotics, including those for the management of SSTIs [39,46,47]. Wiltrakis et al. (2022) demonstrated that a comprehensive set of measures, including refining the STGs, resulted in an appreciable improvement in compliance. For patients with purulent SSTIs, the optimal antibiotic choices and duration increased from a baseline median of 28% up to 64% of patients surveyed, and for non-purulent SSTI, this increased from a median of 2% up to 43% [62]. ASPs have also been successfully introduced in hospitals across Africa, including South Africa, in a variety of circumstances including improving compliance to agreed guidelines and reducing SSIs, providing future guidance [39,46,47,69,70,71]. We will be following this up in this hospital to improve future prescribing, including greater use of culture and sensitivity testing given increasing concerns with resistant organisms. In addition, a closer look at the antibiotics prescribed ensuring, where possible, that those prescribed are mainly “access” antibiotics, with those from the “watch” and “reserve” list requiring justification for their prescribing.

Hospital pharmacists can play a key role in driving forward ASPs in Africa and beyond, building on their success in a number of ASPs across Africa [39,72,73].

We are aware of a number of limitations with this study. Firstly, we only performed the study in one tertiary public hospital and only in the surgical wards for the reasons stated. Secondly, we did not use a validated data collection tool. However, this was based on published papers combined with the considerable experience of the co-authors working in this area. In addition, the forms were piloted to add further robustness to the data collection tools used in practice. Thirdly, we only performed the study over a 3-month period. However, the intention was to identify areas for future ASPs. Despite these limitations, we believe our findings are robust, providing direction for the future.

## 4. Materials and Methods

### 4.1. Study Details

This was a retrospective review of patients with SSTIs in a surgical ward of a leading tertiary hospital in Gauteng Province, South Africa, consisting of approximately 1650 beds. We chose the public healthcare system for this study as it accounts for the vast majority of patients in South Africa [74], and South Africa is moving towards universal healthcare.

The study took place between April to June 2021, and involved all patients who were on antimicrobials and admitted to the surgical wards during the data collection period. The surgical wards were targeted as they were most likely to have a higher prevalence of patients with SSTIs.

The principal researcher (A.B.M.) identified patients with SSTIs from all the patient files in the surgical wards during the collection period. Once identified, pertinent files were put aside for review and analysis. Thereafter, the only exclusion criteria were patients with SSTIs not receiving antibiotics.

### 4.2. Data Collection Form

The files of individuals presenting with SSTIs and prescribed antimicrobials were reviewed and the information was extracted anonymously using an adapted data collection tool (Appendix A) by the principal researcher (A.B.M.).

The data collection tool was based on previous studies and was adapted by the co-authors based on their considerable experience [5,8,16]. This included breaking SSTIs down into uncomplicated and type, as well as complicated and type, building on previous classifications [5,7,8]. We successfully used this methodology in previous studies when developing context-specific data collection forms [75,76,77].

Subsequently, a pilot study was undertaken in one of the internal medicine wards at the study site. Ten files from ten patients were reviewed. The data collection tool was found to be clear, concise, and reliable. Consequently, no changes were made to the data collection tool before it was used in the full study.

The information extracted included patient demographics, clinical data such as the SSTI classification, symptoms upon admission, culture and infective pathogens, and the prescribed antibiotics.

Prescribed antibiotics were subsequently compared with the recommendations in the South African STG/EML in order to evaluate treatment compliance [59]. Prescribed antibiotics were also broken down by the WHO AWaRe classification, with “watch” and “reserve” antibiotics key areas for ASPs to reduce the resistance potential [64,65,78]. The “access” group of antibiotics are considered as generally first- or second-line antibiotics for common or severe clinical syndromes, typically having a narrow spectrum and low resistance potential. The “watch” group of antibiotics have a higher resistance potential as well as side-effects, with the “reserve” group only recommended as a last resort and typically prioritized for ASPs alongside agreed quality indicators [39,64,65,78]. This builds on previous antibiotic utilization research projects undertaken by some of the co-authors in South Africa [17,79]. Lastly, patient files were reviewed until the data collection period was over or until the patient was discharged.

### 4.3. Data Analysis

The data collected at the tertiary hospital were entered onto Microsoft Excel™ (Microsoft^®^ Corporation, Redmond, WA, USA). The data were imported into version 25 IBM SPSS Statistics for Windows (IBM Corp., Armonk, NY, USA), where the data were analyzed and presented using tables, diagrams, and flow charts.

## 5. Conclusions

The most common causative agent for SSTIs is *Staphylococcus aureus*, and empiric treatment should cover this prior to the findings from culture and sensitivity testing. It is important to subsequently perform culture sensitivity and susceptibility in order to de-escalate therapy and fully eradicate the causative microorganism. However, cultures were not routinely performed in this hospital and there was variable compliance with the current South African STG. The inability to routinely perform culture and sensitivity testing led to appreciable non-compliance to the current South African STG in our study and an associated increased risk of AMR. STGs should be routinely followed and adhered to in order to reduce hospital stay, costs, and resistance potential. These can be part of future ASPs in this hospital and beyond in South Africa, with hospital pharmacists playing a key role to ensure the safe and effective use of antibiotics in hospitals. Alongside this, hospital pharmacists should work more closely with microbiologists and surgeons to ensure that prescribing guidelines are routinely followed and that treatment with antibiotics is based primarily upon culture and sensitivity testing. We will be following this up in future studies.

## Figures and Tables

**Figure 1 antibiotics-12-00275-f001:**
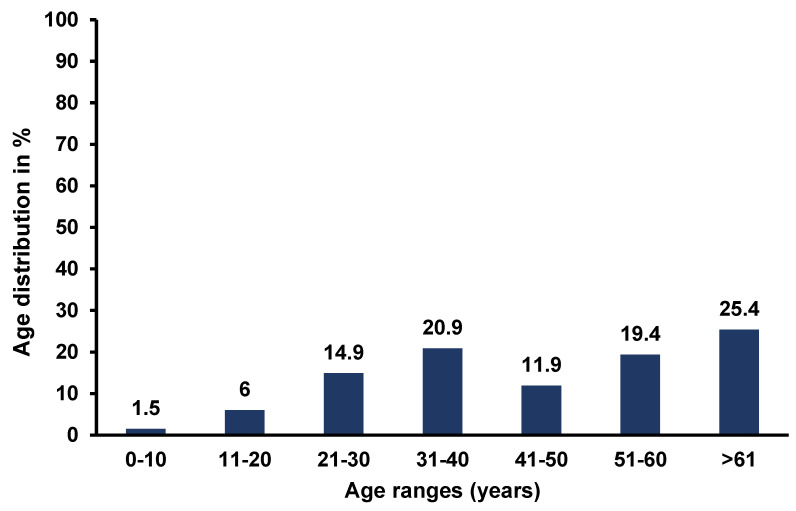
Age distribution of patients with SSTIs.

**Figure 2 antibiotics-12-00275-f002:**
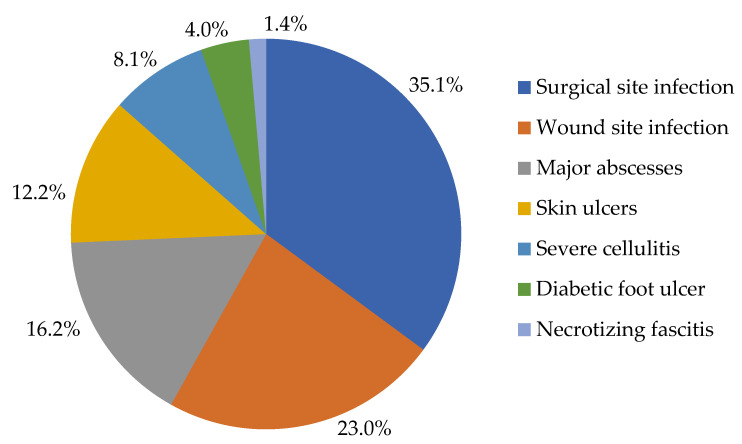
The different types of complicated SSTIs.

**Figure 3 antibiotics-12-00275-f003:**
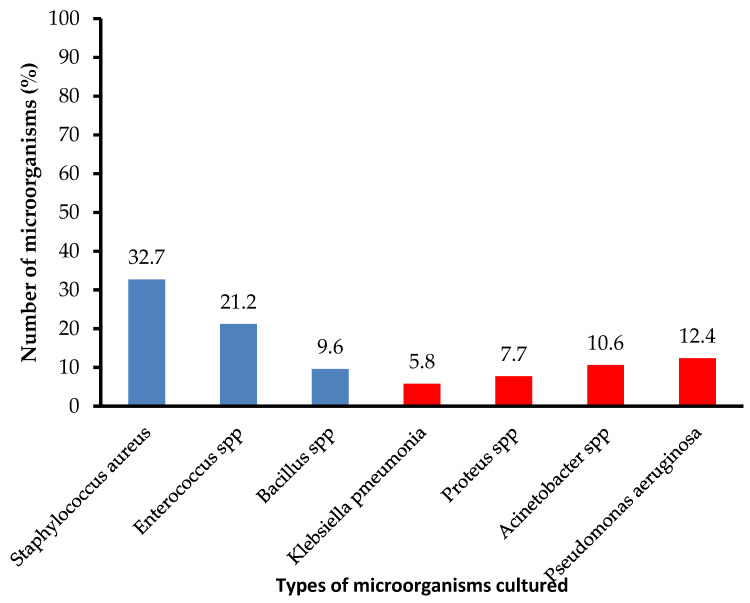
Cultured microorganisms. NB: Blue shaded bars represent Gram-positive microorganisms and the red shaded bars represent Gram-negative microorganisms.

**Table 1 antibiotics-12-00275-t001:** Demographic details and clinical characteristics of the patients that participated in the study.

Demographics and Clinical Characteristics		*n* (67, %)
Gender	Male Female	47 (70.1)
		20 (29.9)
Smoking	Yes	43 (64.2)
	No	24 (35.8)
Alcohol	Yes	46 (68.7)
	No	21 (31.3)
Comorbidities	Hypertension	15 (22.4)
	Diabetes mellitus	6 (9.0)
	Chronic osteomyelitis	9 (13.4)
	HIV positive	8 (11.9)
	Anemia	2 (3.0)
	Depression	2 (3.0)
	None	25 (37.3)
Signs and symptoms	Infection (pus, foul smell)	56 (83.6)
	Inflammation (tenderness, itching)	50 (74.6)
	Skin ulceration (sores, blisters)	23 (34.3)
Types of surgery that resulted in skin infection	Leg surgery	19 (28.4)
	Leg amputation	6 (4.0)
	Skin graft	2 (3.0)
	Spinal surgery	1 (1.5)
	Pelvic surgery	2 (3.0)
	None	39 (58.0)

HIV = Human immunodeficiency virus.

**Table 2 antibiotics-12-00275-t002:** Distribution of microorganisms according to the diagnosis.

Diagnosis	Gram Positive *n* (%)	Gram Negative *n* (%)
Wound site infection	4 (12.1%)	5 (26.3%)
Surgical site infection	9 (27.3%)	5 (26.3%)
Major abscesses	5 (15.2%)	3 (15.8%)
Severe infective cellulitis	4 (12.1%)	2 (10.5%)
Skin ulcers	8 (24.2%)	3 (15.8%)
Diabetic foot ulcer	3 (9.1%)	1 (5.3%)
Total	33 (63.5%)	19 (36.5%)
	52	
More than one pathogen cultured	14 (59.2%)	

**Table 3 antibiotics-12-00275-t003:** The different antibiotics prescribed to treat the various SSTIs.

Antimicrobial	*n* (%)	WHO AWaRe Classification
Cefazolin	25 (31.3%)	Access
Gentamycin	14 (17.5%)	Access
Ciprofloxacin	9 (11.3%)	Watch
Rifampicin	7 (8.8%)	Watch
Cloxacillin	5 (6.3%)	Access
Amoxicillin/clavulanic acid	4 (5.0%)	Access
Meropenem	3 (3.8%)	Watch
Ceftriaxone	3 (3.8%)	Watch
Metronidazole	3 (3.8%)	Access
Vancomycin	1 (1.3%)	Watch
Azithromycin	1 (1.3%)	Watch
Colistin	1 (1.3%)	Reserve
Flucloxacillin	1 (1.3%)	Access
Imipenem	1 (1.3%)	Reserve
Doxycycline	1 (1.3%)	Access
Tazocin	1 (1.3%)	Watch
Total	80 (100%)	

NB: “access”, “watch”, and “reserve” antibiotics taken from the WHO AWaRe list, see Methods.

**Table 4 antibiotics-12-00275-t004:** Microorganisms cultured and their susceptibility.

Antibiotics	Gram Positive *n* (%)				Gram Negative *n* (%)	
	*S. Aureus n = 17*		*Enterococcus n = 12*		*P.aeruginosa n = 7*	
	S	R	S	R	S	R
Ciprofloxacin	4 (24.0)	-	4 (33.0)	-	3 (50.0)	-
Gentamycin	3(18.0)	1(6.0)	6 (50.0)	1 (8.0)	4 (67.0)	-
Cefazolin	5 (29.0)	-	1 (8.0)	-	-	-
Cloxacillin	5 (29.0)	-	1 (8.0)	-	-	-
Amoxicillin	-	-	2 (17.0)	1 (8.0)	-	-
Penicillin	-	2 (12.0)	-	-	-	-

S = sensitive, R = resistance.

**Table 5 antibiotics-12-00275-t005:** Antibiotics given as empiric treatment.

Antibiotic	Number of Patients *n* = 67 (%)	WHO AWaRE Classification
Cefazolin	31 (46.3%)	Access
Amoxicillin/clavulanic acid	20 (29.9%)	Access
Ciprofloxacin	1 (1.5%)	Watch
Amoxicillin None empirically	1 (1.5%) 14 (20.9%)	Access

NB: “access” and “watch” antibiotics taken from the WHO AWaRe list, see Methods.

## Data Availability

Further data are available from the corresponding authors upon reasonable request.

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
