# Peer review of "Antimicrobial Management of Skin and Soft Tissue Infections among Surgical Wards in South Africa: Findings and Implications"

_antibiotics, 2023, doi:10.3390/antibiotics12020275_

Round 1

Reviewer 1 Report

General comments:

This manuscript looks at South Africa Standard Treatment Guidelines (STG) in relationship to treatment of Skin and Soft Tissue Infections (SSTI). It reveals an apparent disconnect between antibiotic prescribing patterns of the STG and skin and soft tissue infections.

The Introduction section draws heavily upon SSTI incidence in the US but does not give similar data for S. Africa in comparison. It should be explicitly stated whether or not similar incidence figures exist for S. Africa. If similar data is unavailable, the authors can propose that this weakness represents an opportunity to gather more data to correct for the relative lack of available information.

The authors suggest that more data is needed to detect overall trends in antimicrobial stewardship in S. Africa. Accordingly, they should describe what studies are needed to help validate, or refute, their initial findings. This may involve collecting data over an extended time frame and/or multi-institutional studies.

Many hospitals have a standing Pharmacy and Therapeutics (P&T) committee that decides which drugs will appear on the facility’s drug formulary. In doing so, they weigh the costs and benefits of each drug and decide which ones are the most effective per dollar (Rand) spent. There may be the need for an occasional variance from provided treatment guidelines. Consider adding this information to a discussion of potential corrective measures.

Based upon their conclusions, the authors should suggest that a closer partnership is needed between pharmacists and microbiology lab personnel. This helps to ensure that prescribing guidelines are routinely followed for empiric antibiotic therapy and that definitive therapy is based primarily upon culture & sensitivity (C&S) testing. C&S testing represents an increased short-term cost but is well worth it in the long run to keep antimicrobial resistance from quickly developing in a patient population.

Specific comments:

Line 26 - Expand on the term “pertinent patients” as to exclude certain populations mentioned later (post-partum OB/GYN) not specifically mentioned.

Lines 31 - Capitalize genus names of “Staphylococcus aureus and Enterococcus spp”. Remove italics from spp.

Line 35 - Change from “staphylococcus” toStaphylococcus”.

Line 46 - Change “layer” to plural form “layers” or to “… at either the epidermal or dermal layers”.

Line 48 - Change “as complex” to “as either complex”.

Lines 49-53 - Define (differentiate) between uncomplicated and complicated SSTI.

Line 50 - Change “having a” to “being”.

Line 55 - Change “Testing” to Rapid testing”

Lines 59-60 - The overall incidence of SSTIs in S. Africa should be included for comparison to US data. If unknown, it should be stated as such.

Line 59-68 - Incidence figures should be similarly provided for all stated conditions (e.g., pneumonia, parenteral drug users, etc.) since references are indicated.

Lines 70-71 - Is the cost for treating SSTIs in S. Africa known?

Line 74 - Change “Streptococci” to either “streptococci” or “Streptococcus spp.”

Line 76 - Change “Whilst bacterial organisms” to “While bacteria”.

Line 78 - Indicate that rapidly growing mycobacteria (RGM) are non-tuberculous mycobacteria.

Line 80 - Indicate if effective management listed is for complicated SSTIs.

Line 84 - Change “antibiotic resistance (AMR)” to “antimicrobial resistance (AMR)”.

Line 91 - Include the following reference about antimicrobial stewardship:

Chetty S, Reddy M, Ramsamy Y, Naidoo A, Essack S. Antimicrobial stewardship in South Africa: a scoping review of the published literature. JAC Antimicrob Resist. 2019 Nov 28;1(3):dlz060. doi: 10.1093/jacamr/dlz060. PMID: 34222934; PMCID: PMC8210007.

Line 94 - Why has adherence to the guidelines been variable in S. Africa? What factors contribute to this effect?

Line 127 - The authors need to state if anaerobic cultures were likewise accomplished on submitted samples. Characterizing samples as having a “foul smell” may indicate the presence of anerobic bacteria (e.g., Clostridium perfringens).

Line 133 - The 25% figure listed for wound infections does not match the 23% depicted on Fig 2.

Table 2 - Table information needs to be further broken down to either gram-positive cocci or gram-positive rods. The same goes for gram-negative isolates. Additionally, Table 2 states 63.5%  gram-positive but Fig 3 gram-positive isolates add up to 65.4%. The difference between these 2 items needs explained.

Figure 3 - Bacillus is misspelled.

Line 170 - Change to “A total of 16 different antibiotics were prescribed to patients with SSTIs”.

Lines 174-75 - Define WHO AWaRe criteria for Access, Watch, Reserve here or in Table 3 legend.

Table 4 - Change table legend from “S= sensitivity” to S= sensitive”.

Line 191 - Change “given as treatment” to “given as definitive treatment”.

Line 218 - Elaborate some on the statement “this may be due to local factors.”

Line 233 - HIV also needs to listed as a comorbidity in the introduction section , if it’s being included in the discussion section.

Lines 240-41 - Capitalize “staphylococcus”. Same goes for “proteus” and “klebsiella” in the next two lines.

Lines 256-57 - Specify empiric treatment.

Lines 358 - Capitalize “staphylococcus”.

Author Response

General comments:

1) This manuscript looks at South Africa Standard Treatment Guidelines (STG) in relationship to treatment of Skin and Soft Tissue Infections (SSTI). It reveals an apparent disconnect between antibiotic prescribing patterns of the STG and skin and soft tissue infections.

Author comments: Thank you for this.

2) The Introduction section draws heavily upon SSTI incidence in the US but does not give similar data for S. Africa in comparison. It should be explicitly stated whether or not similar incidence figures exist for S. Africa. If similar data is unavailable, the authors can propose that this weakness represents an opportunity to gather more data to correct for the relative lack of available information.

Author comments: Thank you for this. We drew principally on data from the US to highlight that this is a priority area since most data on incidence rates has been published in the US. Unfortunately, as mentioned, there is currently limited data across Africa/ South Africa on current incidence rates. However, we have documented some of these. Consequently, this is why we believe papers such as this are so important. We hope you agree.

3) The authors suggest that more data is needed to detect overall trends in antimicrobial stewardship in S. Africa. Accordingly, they should describe what studies are needed to help validate, or refute, their initial findings. This may involve collecting data over an extended time frame and/or multi-institutional studies.

Author comments: Thank you. We have now added to this in the revised paper. In fact, we have now published in Antibiotics a paper documenting utilisation patterns for antimicrobials across Africa including ASPs – which have just started (New Ref 39). This builds in the papers of Akpan et al and Siachalinga et al (New refs 46 and 47). We have also included your suggested paper (new Ref 52) as well as the paper of Engler et al which shows pro-activity in South Africa to implement ASPs (new Ref 40) as well as examples – building on Chetty et al (Ref 52). These examples are also documented in new Ref 39 (extensive publication on PPS/ ASP studies across Africa) . We hope this is now acceptable.

4) Many hospitals have a standing Pharmacy and Therapeutics (P&T) committee that decides which drugs will appear on the facility’s drug formulary. In doing so, they weigh the costs and benefits of each drug and decide which ones are the most effective per dollar (Rand) spent. There may be the need for an occasional variance from provided treatment guidelines. Consider adding this information to a discussion of potential corrective measures.

Author comments: Thank you. As I am sure you are aware, P & T committees are variable across Africa even among tertiary hospitals in some countries (e.g. Nigeria - quoted). In addition – seeing the development of IPCs (discussed – new refs 39, 46-49). The key area is the instigation of active ASP groups within hospitals with hospital pharmacists playing a key role. This is now happening across Africa (as seen in e.g. Ref 39). We have now expanded on this in the updated paper, and hope this is now acceptable.

5) Based upon their conclusions, the authors should suggest that a closer partnership is needed between pharmacists and microbiology lab personnel. This helps to ensure that prescribing guidelines are routinely followed for empiric antibiotic therapy and that definitive therapy is based primarily upon culture & sensitivity (C&S) testing. C&S testing represents an increased short-term cost but is well worth it in the long run to keep antimicrobial resistance from quickly developing in a patient population.

Author comments: Thank you – we have now refined the conclusion and hope this is now acceptable.

Specific comments:

1) Line 26 - Expand on the term “pertinent patients” as to exclude certain populations mentioned later (post-partum OB/GYN) not specifically mentioned.

Author comments: Thank you – now addressed. We hope this is now acceptable.

2) Lines 31 - Capitalize genus names of “Staphylococcus aureus and Enterococcus spp”. Remove italics from spp.

Author comments: Thank you – now done

3) Line 35 - Change from “staphylococcus” to “Staphylococcus”.

Author comments: Thank you now done

4) Line 46 - Change “layer” to plural form “layers” or to “… at either the epidermal or dermal layers”.

Author comments: Thank you – now done

5) Line 48 - Change “as complex” to “as either complex”.

Author comments: Thank you – now done

6) Lines 49-53 - Define (differentiate) between uncomplicated and complicated SSTI.

Author comments: Thank you – now updated. We hope this is now acceptable.

7) Line 50 – Change “having a” to “being”.

Author comments: Thank you – now done

8) Line 55 - Change “Testing” to Rapid testing”

Author comments: Thank you – now done

9) Lines 59-60 - The overall incidence of SSTIs in S. Africa should be included for comparison to US data. If unknown, it should be stated as such.

Author comments: Thank you. As stated earlier – no such data currently exists in South Africa or generally across Africa. This will change with greater implementation of the NAPs, etc. In addition, we are hoping that this manuscript (if and when accepted for publication and subsequently published) will prompt more research on SSTIs in South Africa as well as other African countries. We hope this is now OK.

10) Line 59-68 - Incidence figures should be similarly provided for all stated conditions (e.g., pneumonia, parenteral drug users, etc.) since references are indicated.

Author comments: Thank you for this, However, as stated earlier there is limited data on this outside of the USA. However – we have added more data where we can from the US studies. We hope this is now acceptable.

11) Lines 70-71 - Is the cost for treating SSTIs in S. Africa known?

Author comments: Thank you for this comment. Currently, there is very little information generally regarding costings among public hospitals in South Africa. We are trying to address this generally and for specific areas.

12) Line 74 - Change “Streptococci” to either “streptococci” or “Streptococcus spp.”

Author comment: Thank you – now done

13) Line 76 - Change “Whilst bacterial organisms” to “While bacteria”.

Author comment: Thank you – now done

14) Line 78 - Indicate that rapidly growing mycobacteria (RGM) are non-tuberculous mycobacteria.

Author comment: Thank you. However, now removed this as one Reviewer asked us to cut down on the Introduction where we could. We hope this is acceptable.

15) Line 80 - Indicate if effective management listed is for complicated SSTIs.

Author comments: Thank you - now done.

16) Line 84 - Change “antibiotic resistance (AMR)” to “antimicrobial resistance (AMR)”.

Author comment: Thank you – now done

17) Line 91 - Include the following reference about antimicrobial stewardship:

Chetty S, Reddy M, Ramsamy Y, Naidoo A, Essack S. Antimicrobial stewardship in South Africa: a scoping review of the published literature. JAC Antimicrob Resist. 2019 Nov 28;1(3):dlz060. doi: 10.1093/jacamr/dlz060. PMID: 34222934; PMCID: PMC8210007.

Author comment: Thank you - Now done (new Ref 52). This is in association with more general discussions of ASPs and their impact in Africa including South Africa (with references). We hope this is now OK.

18) Line 94 - Why has adherence to the guidelines been variable in S. Africa? What factors contribute to this effect?

Author comments: Thank you – we have now expanded on this section with additional references, and hope this is now acceptable.

19) Line 127 - The authors need to state if anaerobic cultures were likewise accomplished on submitted samples. Characterizing samples as having a “foul smell” may indicate the presence of anerobic bacteria (e.g., Clostridium perfringens).

Author comments: Thank you - No anaerobic cultures were observed or found during the research. We have now included this in the revised paper, and hope this is now acceptable.

20) Line 133 - The 25% figure listed for wound infections does not match the 23% depicted on Fig 2.

Author comments: Thank you – now corrected to 23%.

21) Table 2 - Table information needs to be further broken down to either gram-positive cocci or gram-positive rods. The same goes for gram-negative isolates. Additionally, Table 2 states 63.5%  gram-positive but Fig 3 gram-positive isolates add up to 65.4%. The difference between these 2 items needs explained.

Author comments: Thank you for this – now revised. We hope this is now OK.

22) Figure 3 - Bacillus is misspelled.

Author comments: Thank you - now updated

23) Line 170 - Change to “A total of 16 different antibiotics were prescribed to patients with SSTIs”.

Author comment: Thank you – now updated.

24) Lines 174-75 - Define WHO AWaRe criteria for Access, Watch, Reserve here or in Table 3 legend.

Author comment: Thank you – now done in the legend and in the updated Methodology. We hope this is now acceptable.

25) Table 4 - Change table legend from “S= sensitivity” to S= sensitive”.

Author comment: Thank you – now done

26) Line 191 - Change “given as treatment” to “given as definitive treatment”.

Author comment: Thank you – now done

27) Line 218 - Elaborate some on the statement “this may be due to local factors.”

Author comments: Thank you – now inserted.

28) Line 233 - HIV also needs to listed as a comorbidity in the introduction section , if it’s being included in the discussion section

Author comments: Thank you – now mentioned in the Discussion. We hope this is now OK.

29) Lines 240-41 - Capitalize “staphylococcus”. Same goes for “proteus” and “klebsiella” in the next two lines.

Author comment: Thank you – now done

30) Lines 256-57 - Specify empiric treatment.

Author comments: Thank you - now updated. We hope this is now acceptable.

31) Lines 358 - Capitalize “staphylococcus”.

Author comment: Thank you – now done

Reviewer 2 Report

I found this article interesting for the readers and followed the journal Antibiotics’ scope. I don’t have any major comments as this article has enough data and is well written with proper discussion.

I would recommend the article be published in Antibiotics after minor corrections. 

The author needs to address the following comments/corrections.

1.     The author should correct the format of references wherever needed (e.g Year Bold, Volume Italic etc).

2.     Introduction needs to be shortened.

3.     The author could have a table having antimicrobial used for Skin and soft tissue (SSTIs) infections with comorbidities.

4.     The author could have discussed a bit more about antimicrobial resistance Skin and soft tissue (SSTIs) infections.

Author Response

Comments and Suggestions for Authors

I found this article interesting for the readers and followed the journal Antibiotics’ scope. I don’t have any major comments as this article has enough data and is well written with proper discussion.

Author Comments: Thank you for your positive comments – appreciated!

I would recommend the article be published in Antibiotics after minor corrections. The author needs to address the following comments/corrections.

Author comments: Thank you for these. We hope we have adequately addressed these.

  1. The author should correct the format of references wherever needed (e.g Year Bold, Volume Italicetc).

Author comments: Thank you – the references will be updated if needed during the publication stage.

  1. Introduction needs to be shortened.

Author comments:  Thank you – we have now shortened this where we can. However, another reviewer asked for additional data in the Introduction to improve this – certainly around incidence figures outside of USA and in Africa/ South Africa as well as issues such as ASPs, etc. We hope this is acceptable.

  1. The author could have a table having antimicrobial used for Skin and soft tissue (SSTIs) infections with comorbidities.

Author comments: Thank you. We have tried looking at this – but this proved difficult. We have though looked at co-morbidities further in South Africa including HIV – and have listed this as a possible reason (with others) for differences in the findings between countries. We hope this is acceptable.

  1. The author could have discussed a bit more about antimicrobial resistance Skin and soft tissue (SSTIs) infections.

Author comments: Thank you for this comment. We have now extensively revised the paper to discuss more about enhancing the appropriate use of antibiotics not just for SSTIs but wider through active ASPs. ASPs were seen as a problem initially in LMICs including South Africa due to resource issues (finances and personnel). However – this is changing especially in South Africa and we have provided a number of references to support this statement – with South Africa leading the way in Africa. We hope this is now acceptable.

Round 2

Reviewer 1 Report

The genus Bacillus appears to be still misspelled in Figure 3.

Author Response

Comments and Suggestions for Authors

The genus Bacillus appears to be still misspelled in Figure 3.

Author comment: Thank you – now addressed.
